# Molecular Mechanisms Underlying Ca^2+^/Calmodulin-Dependent Protein Kinase Kinase Signal Transduction

**DOI:** 10.3390/ijms231911025

**Published:** 2022-09-20

**Authors:** Hiroshi Tokumitsu, Hiroyuki Sakagami

**Affiliations:** 1Applied Cell Biology, Graduate School of Interdisciplinary Science and Engineering in Health Systems, Okayama University, Okayama 700-8530, Okayama, Japan; 2Department of Anatomy, Kitasato University School of Medicine, Sagamihara 252-0374, Kanagawa, Japan

**Keywords:** CaMKK, CaM-kinase cascade, Ca^2+^ signaling, phosphorylation

## Abstract

Ca^2+^/calmodulin-dependent protein kinase kinase (CaMKK) is the activating kinase for multiple downstream kinases, including CaM-kinase I (CaMKI), CaM-kinase IV (CaMKIV), protein kinase B (PKB/Akt), and 5′AMP-kinase (AMPK), through the phosphorylation of their activation-loop Thr residues in response to increasing the intracellular Ca^2+^ concentration, as CaMKK itself is a Ca^2+^/CaM-dependent enzyme. The CaMKK-mediated kinase cascade plays important roles in a number of Ca^2+^-dependent pathways, such as neuronal morphogenesis and plasticity, transcriptional activation, autophagy, and metabolic regulation, as well as in pathophysiological pathways, including cancer progression, metabolic syndrome, and mental disorders. This review focuses on the molecular mechanism underlying CaMKK-mediated signal transduction in normal and pathophysiological conditions. We summarize the current knowledge of the structural, functional, and physiological properties of the regulatory kinase, CaMKK, and the development and application of its pharmacological inhibitors.

## 1. Introduction

Ca^2+^/calmodulin-dependent protein kinase (CaMK) is a Ser/Thr kinase activated by binding of a versatile Ca^2+^-signal transducer, calmodulin (CaM), to various extracellular stimuli, including hormones, neurotransmitters, etc., resulting in an increasing intracellular Ca^2+^ concentration [1,2]. Similar to other protein kinases, CaMK phosphorylates specific residue(s) in certain cellular proteins regulating functions such as enzymatic efficiency, cytoskeletal organization, transcriptional regulation, and receptor activity. Based on their substrate specificity, these enzymes can be classified into two groups: enzymes with limited physiological functions and multifunctional CaMKs with a broad substrate specificity. The former includes myosin light chain kinase (MLCK) and phosphorylase kinase γ-subunit, which specifically phosphorylate myosin light chain for Ca^2+^-regulated smooth muscle contraction and phosphorylase b for glycogen degradation, respectively [3,4,5]. On the other hand, multifunctional CaMKs, including CaMKI, CaMKII, and CaMKIV, can phosphorylate multiple cellular proteins to transduce Ca^2+^ singling to cellular physiology [6]. Regardless of CaMKs’ substrate specificity, their molecular structure is similar. An amino terminal catalytic domain is followed by a regulatory domain containing an autoinhibitory segment overlapping with the CaM binding sequence [7,8,9,10]. Basically, the catalytic activity of CaMK is suppressed by the interaction with its own autoinhibitory region under low intercellular Ca^2+^ concentrations. Upon stimulation with extracellular signals that increase intracellular Ca^2+^ concentrations, CaM binds Ca^2+^ ions, which induces conformational changes to interact with the regulatory domain of CaMKs. Ca^2+^/CaM binding allosterically changes the autoinhibition state of the CaMK catalytic domain to the active state. In addition to Ca^2+^/CaM-binding, some CaMKs are activated by phosphorylation, either autophosphorylation or trans-phosphorylation, by activating kinases. Numerous studies have demonstrated that the CaMKII holoenzyme (10~12 kinases) undergoes intermolecular autophosphorylation at Thr286 (CaMKIIα) in the autoinhibitory domain, generating an autonomous activity (50%~80% of total activity) even in the absence of the activator, Ca^2+^/CaM [10]. This unique enzymatic feature could partly explain the various biological reactions induced by transient Ca^2+^-signaling, such as long-term synaptic plasticity, related to memory and learning [11]. Other CaMKs, such as CaMKI and CaMKIV, are monomeric enzymes localized primarily in the cytoplasm and nuclei, respectively [12,13,14]. In addition to enzymatic activation by Ca^2+^/CaM binding, both CaMKs are phosphorylated at activation-loop Thr residues (Thr177 in CaMKIα and Thr196 in CaMKIV) in the catalytic domain by an upstream kinase, CaMK-kinase (CaMKK), resulting in significant catalytic activation [1,2]. Indeed, CaMKK-mediated CaMKI and CaMKIV activation has been shown to regulate various cellular processes, including neuronal morphogenesis, synaptic plasticity, and transcriptional activation through phosphorylation of transcription factors such as the cAMP-response element binding protein (CREB). Therefore, the signal transduction system mediated by the Ca^2+^-dependent kinase cascade has been called the “CaMK cascade.” This review summarizes the present knowledge on CaMKK, including the enzyme activation, cellular localization, target kinases, low-molecular weight inhibitors, and signaling pathways, as well as its putative physiological and pathophysiological functions based on in vitro and in vivo analyses.

## 2. Discovery and Current Members of CaMKK Family

In 1993, CaMKK activity was first detected in a rat brain extract as a CaMKIV-activating activity [15]. Soon after, a CaMKIV-activating kinase with a molecular mass of 66–68 kDa on sodium dodecyl sulfate–polyacrylamide gel electrophoresis (SDS-PAGE) was purified from rat brain extracts by multiple column chromatography, and its CaMKIV-activating activity was detected by denaturation/renaturation in an SDS-PAGE gel [16,17]. In 1994, Lee and Edelman purified a 53-kDa CaMKIa activator from the porcine brain [18]. The characteristic features of the CaMKIV-activating kinase and the CaMKIa activator, including the Ca^2+^/CaM binding ability and protein kinase activity, were quite similar [19]. In 1995, we successfully cloned the complete cDNA for rat brain 68-kDa CaMKIV-activating kinase based on partial amino acid sequences from the purified enzyme [20]. Based on the deduced amino acid sequence, this 68-kDa CaMKIV-activating kinase was composed of 505 residues and contained an N-terminal catalytic domain similar to other CaMKs and cAMP-dependent protein kinase (PKA) (Table 1). Transfecting this cDNA in COS-7 cells showed it expressed a 68-kDa CaM binding protein capable of activating both recombinant CaMKIV and CaMKI. Thus, we called the enzyme CaMKK. Later, two groups independently cloned another CaMKK isoform (CaMKKβ or CaMKK2) from rat and human cDNAs encoding 587 and 588 amino acid residues, respectively, with ~70% amino acid sequence identity with the first identified isoform [21,22]. Accordingly, the 68-kDa CaMKK isoform was denoted as CaMKKα or CaMKK1. Hsu et al. and our lab reported the genomic organization and transcription of the human CaMKKβ/2 gene and two functional splicing variants (β-2 encoding 533 amino acids and β-3 encoding 541 amino acids) in addition to the originally cloned CaMKKβ-1 encoding 588 amino acids [23,24] (Table 1). All human CaMKKβ/2 splice isoforms (β1–3) contain an identical N-terminal region including a catalytic domain (residues 165–419) and a regulatory domain (residues 475–500) with distinct C-terminal regions; however, the functional differences and/or redundancy of the isoforms remain unclear. 

Since the first cloning of CaMKKα/1 from rats, CaMKK orthologues have been identified and their functions have been evaluated in various eukaryotic species. In the nematode *Caenorhabditis elegans*, a CaMKK orthologue with 432 amino acid residues (CKK-1) was identified as a CaMK that phosphorylates the activation-loop Thr179 in *C. elegans* CaMKI/IV (CMK-1) [38,40] (Table 1). The CKK-1/CMK-1 cascade regulates CREB (CRH-1)-dependent transcription in a subset of head neurons in living nematodes and in vitro [38]. Furthermore, temperature-related changes in the gene expression are mediated by CMK-1 in AFD sensory neurons [44]. During heat acclimation, CKK-1-dependent phosphorylation of CMK-1 controls CMK-1 translocation into the nucleus to reduce thermal avoidance [45]. Joseph and Means identified a CaMKK orthologue (CMKC) in *Asperugillus nidulans* encoding 518 amino acid residues with ~30% sequence identity with rat CaMKK, which phosphorylates and increases the activity of CMKB, a CaMKI/IV orthologue [41]. The disruption of *cmkc* in *A. nidulans* revealed that CMKC is important for proper timing of the first nuclear division after germination, similar to the downstream kinase, CMKB. Based on its amino acid sequence similarity with mammalian CaMKK, the putative CaMKK Ssp1 was identified in fission yeast *Schizosaccharomyces pombe* and has been reported to control G2/M transition and response to stress [42], although the Ca^2+^/CaM-dependency of Ssp1 activity has not been demonstrated. Ssp1 was shown to activate the catalytic subunit of AMPK (Ssp2) through phosphorylation at Thr189 in the activation-loop, resulting in redistribution of the fission yeast AMPK orthologue from the cytoplasm to the nucleus. This pathway is important for the correct transition from cell proliferation to cell differentiation under low-energy conditions [43].

## 3. Tissue Distribution and Subcellular Localization of CaMKK

Both CaMKKα/1 and CaMKKβ/2 are expressed most abundantly in the rodent brain, both at the mRNA and protein levels (Figure 1A and Figure 2A). In the rat brain, CaMKKα/1 and CaMKKβ/2 show distinct but overlapping gene expression patterns, as revealed by in situ hybridization [46]. Both their mRNAs are expressed at various levels in the olfactory bulb, piriform cortex, olfactory tubercle, hippocampal formation, cerebral cortex, tenia tecta, striatum, cerebral granular layer, and spinal dorsal horn. Generally, CaMKKα/1 mRNA is more widely expressed in the neuronal nuclei in the diencephalon, brain stem nuclei, and spinal cord. Although the expression pattern of AMPK mRNA in the brain is unknown, the expression patterns of CaMKKα/1 and CaMKKβ/2 in the brain appear to closely resemble those of CaMKI and CaMKIV, respectively. In addition, CaMKKs are expressed in various non-neural tissue and cell types, but at relatively low levels compared with the brain. Indeed, a faint but discrete CaMKKα/1 mRNA expression was detected in the thymus and spleen, in addition to its abundant signals in the brain [20]. Quantitative RT-PCR analysis of CaMKKβ/2 in mouse tissues revealed ≥10 times lower CaMKKβ/2 expression levels in non-neural tissues, including white and brown adipose tissues, as well as heart, kidney, liver, lung, and muscle, compared with in the brain [47]. The expression of CaMKKα/1 in non-neural tissues other than rat insulin-producing pancreatic β cells remains unclear [48]. On the other hand, CaMKKβ/2 is expressed at mRNA and/or protein levels in the mouse preadipocytes [49], fibroblasts [49], hepatocytes [50], monocytes [51], macrophages [51], hematopoietic, and mesenchymal stem and progenitor cells in the bone marrow [52,53,54], osteoblasts, and osteoclasts differentiated in vitro from bone marrow cells [52]. The expression of CaMKKβ/2 was also observed in pancreatic α and β cells [55], skeletal muscle cells [47], vascular myocytes (smooth muscle cells) [56], human umbilical cord vein endothelial cells [57], and human adrenal cortical cells in the zona glomerulosa and zona fasciculata [58]. CaMKKβ/2 expression is dynamically regulated during the development of certain cell lineages. First, CaMKKβ/2 mRNA is expressed in common myeloid progenitor and granulocyte–monocyte progenitor cells, but is sharply down-regulated (>30 times) during terminal granulocytic differentiation [54]. Genetic ablation of CaMKKβ/2 in mice results in enhanced granulocytic differentiation in the bone marrow, suggesting that CaMKKβ/2 negatively regulates granulopoiesis [54]. Second, CaMKKβ/2 is expressed abundantly at mRNA and protein levels in primary preadipocytes isolated from mouse white adipose tissue, but is markedly decreased in mature adipocytes [49]. CaMKKβ/2 null mice showed enhanced adiposity with increased adipocyte size and number, suggesting that CaMKKβ/2 negatively regulates adipogenesis [49]. Last, CaMKKβ/2 mRNA and protein levels progressively decrease in the mouse skeletal muscle during postnatal development [47]. Knocking down CaMKKβ/2 promotes the proliferation and differentiation of C2C12 myoblast cells, whereas CaMKKβ/2 overexpression has the opposite effects, suggesting that CaMKKβ/2 negatively regulates myogenesis [47]. These findings suggest that CaMKKβ/2 expression is tightly associated with the maintenance of undifferentiated states and the restriction of fate commitment in stem cells and progenitor cells of certain cell lineages.

Although the immunohistochemcial distribution of CaMKKα/1 and CaMKKβ/2 in the brain is generally consistent with their gene expression patterns described above (Figure 1A and Figure 2A), the subcellular localization of CaMKKα/1 and CaMKKβ/2 in neurons is still under debate. Two independent groups reported somewhat inconsistent immunohistochemical results in the rat brain: Fujisawa’s group, using polyclonal antibodies, demonstrated that CaMKKα/1 was localized exclusively to the nuclei of virtually all central neurons [59], whereas CaMKKβ/2 was localized to both the cytoplasm and nucleus at varying ratios, depending on the neuronal cell types [60]. On the other hand, using monoclonal antibodies, we demonstrated that both CaMKKα/1 and CaMKKβ/2 were localized primarily to the perikaryal cytoplasm and dendrites in most immunoreactive neurons [61]. In sharp contrast with Fujisawa’s findings, CaMKKα/1 was clearly excluded from the nucleus. Despite the extremely low nuclear staining for CaMKKβ/2, its nuclear exclusion was less evident than that of CaMKKα/1. The reasons for this discrepancy remain unknown, but might be explained by different experimental conditions such as fixation, antibody sensitivity, and undefined stimuli, inducing CaMKKs’ translocation or degradation during sample preparation. Therefore, the immunohistochemical localization of CaMKK isoforms remains to be re-examined using identical antibodies under the same conditions. Interestingly, both groups found only cytoplasmic staining for CaMKKβ/2 in cerebellar granule cells [61], where CaMKIV is expressed mostly abundantly in the nucleus [62]. The discrepancy in subcellular localization between CaMKKβ/2 and CaMKIV in some neurons suggests the possibility of Ca^2+^-induced nuclear translocation of CaMKKs or CaMKIV. Accordingly, chronic spiking blockage in the cultured cortical neurons by tetrodotoxin induced nuclear translocation of CaMKKβ/2 and activation of nuclear CaMKIV, thereby regulating alternative splicing of the BK channel through the phosphorylation and nuclear exclusion of Nova-2, an RNA binding protein involved in alternative mRNA splicing [63].

## 4. Domain Structure and Activation of CaMKK

Both mammalian CaMKK isoforms (α/1 and β/2) have been identified as Ca^2+^/CaM binding kinases [20,22]. Similar to other CaMKs, CaMKK is composed of an N-terminal catalytic domain followed by a regulatory domain containing an autoinhibitory segment and a Ca^2+^/CaM binding sequence. Thus, the kinase activity of CaMKKα/1 is strictly regulated by an autoinhibitory mechanism, i.e., the regulatory domain (residues 438–463) blocks the catalytic domain to inhibit the kinase activity; this inhibition is released by Ca^2+^/CaM binding to the C-terminal region of the regulatory domain (Figure 1B) [64]. In the regulatory domain, Ile441 is particularly important for autoinhibition and is conserved in humans, rats, and *C. elegans* (Figure 1C). NMR spectroscopy analysis of Ca^2+^/CaM complexed with the CaMKKα/1 regulatory domain (residues 438–463) peptide revealed that the N- and C-terminal hydrophobic pockets of CaM anchor Trp444 and Phe459 of the CaMKKα/1 peptide, respectively (Figure 1E) [66], in the opposite orientation to other known Ca^2+^/CaM complexes such as CaMKII [69] or MLCK [70,71]. When replacing the Ca^2+^/CaM binding sequence in CaMKKα/1 by that in rat CaMKIIα, rabbit skeletal muscle MLCK, or chicken smooth muscle MLCK, all chimeric CaMKK mutants, exhibited Ca^2+^/CaM-dependent activity like wild type CaMKK, indicating that CaM binding orientation is not critical for releasing CaMKK autoinhibition [64]. This 14-residue separation between two key hydrophobic groups in the regulatory domain is unique among previously determined CaM complexes (Figure 1C,E) [72]. These characteristic features of the Ca^2+^/CaM binding complex with the CaMKK peptide were also observed with the CaM binding peptide (residues 331–356) of *C. elegans* CaMKK (CKK-1) in X-ray crystallography [73]. In contrast with CaMKKα/1, another CaMKK isoform (CaMKKβ/2) is constitutively active, exhibiting a significant Ca^2+^/CaM-independent activity (60–70% of total activity), attributable to the N-terminal regulatory segment (129–151) because a deletion of the N-terminal segment (residues 129–151) from rat CaMKKβ/2 significantly reduces its Ca^2+^/CaM-independent activity (10% of total activity) without any effect on the Ca^2+^/CaM-dependent activity (Figure 2B,C) [22,31]. Although CaMKK α/1 and β/2 had been considered monomeric kinases similar to other CaMKs including CaMKI and CaMKIV, Ling et al. recently reported that FLAG-tagged CaMKKβ/2 and HA-tagged CaMKK2 β/2 mutant (Arg311Cys) might form a dimer or larger oligomer [74]. Consistent with this possibility, we demonstrated the oligomerization of both rat CaMKK isoforms in transfected cells by chemical crosslinking [75]. The CaMKKα/1 oligomer was catalytically active, although the mechanism and functional consequences of CaMKK oligomerization remain unknown.

## 5. CaMKK Signaling Pathway

In addition to CaMKI (at Thr177 in CaMKIα [26]) and CaMKIV (at Thr196 in mouse CaMKIV [19,25]), PKB/Akt is phosphorylated at Thr308 and is activated by CaMKKα/1 in NG108 neuroblastoma cells [27] and LNCaP prostate cancer cells [80], thereby protecting the cells from apoptosis (Table 1 and Figure 3). PKB/Akt phosphorylation by CaMKKβ/2 was observed in ovarian cancer cell lines [32]. In zebrafish, CaMKK stimulates ionocytes or Na^+^-K^+^-ATPase-rich (NaR) cell reactivation via PKB/ Akt activation [81]. In 2005, three independent groups reported that CaMKKβ/2 phosphorylates the catalytic subunit of AMPK (AMPKα) at Thr172, resulting in large enzymatic activation in cultured cells (Figure 3) [33,34,35]. Extensive studies have shown that the CaMKKβ/2-AMPK axis is involved in numerous metabolic and pathophysiological pathways, including cancers and metabolic disorders [82]. SAD-B (BRSK1), a member of the AMPΚ-related family of protein kinases, is phosphorylated at Thr189 and is activated at ~60-fold by CaMKKα/1 in vitro, but not effectively by CaMKKβ/2, even though SAD-B (BRSK1) appears to be a poor substrate for CaMKKα/1 [28,29,83]. CaMKK exhibits a relatively narrow substrate specificity based on the findings of CaMKK’s phosphorylation of activation-loop Thr residues only in a limited number of target kinases. Okuno et al. reported that 5 min heat treatment of CaMKI and CaMKIV at 60°C abolished the phosphorylation by CaMKKα/1, suggesting that native conformations of CaMKI and CaMKIV were necessary for phosphorylation by CaMKKα. Furthermore, the *Km* values for CaMKI and CaMKIV are approximately 1 µM, two to three orders of magnitude lower than that for a CaMKIV peptide substrate (KKKK-189EHQVLMKTVCGTPGY203) containing Thr196 [84], indicating that CaMKK preferably recognizes the tertiary structure of the target kinases rather than the primary amino acid sequence around the phosphorylation Thr residue. According to the amino acid sequence comparison of CaMKK with various protein kinases, CaMKK contains a unique segment with Arg/Pro rich 23 amino acid residues (RP domain) between kinase subdomain II and III (Figure 1D) [20]. RP-domain deletion from CaMKKα’s catalytic domain impaired CaMKI and CaMKIV, but not PKB/Akt phosphorylation and activation, suggesting that the RP domain is involved in CaMKI and CaMKIV recognition as substrates [39]. A further study confirmed the requirement of the RP domain of both CaMKK isoforms for CaMKI, CaMKIV, and AMPK interaction and phosphorylation in vitro [65]. Interestingly, RP-sequence insertion between kinase subdomains II and III of the catalytic domain of liver kinase B1 (LKB1), an alternative activating kinase for AMPK incapable of phosphorylating CaMKI and CaMKIV, resulted in the acquisition of the CaMKIα- and CaMKIV-phosphorylating activity in the LKB1 mutants. This strongly indicates that CaMKK specifically recognizes and phosphorylates CaMKI, CaMKIV, and AMPK through the RP domain; however, this needs to be confirmed by further structural studies considering the RP domain/substrate interaction. Note that non-kinase substrates, including Syndapin I at Thr355 (by both isoforms) [30], SIRT1 at Ser27 and Ser47 [36], GAPDH, and Pex3 (by β/2) [37], were also shown to be phosphorylated by CaMKK in vitro (Table 1).

To date, the upstream CaMKK-activating kinase remains unknown; however, CaMKK is regulated by phosphorylation via multiple protein kinases, as well as autophosphorylation. The first report on CaMKK phosphorylation revealed that CaMKKα/1 was phosphorylated at Thr108 and Ser458 by PKA in transfected COS-7 cells, PC12 cells, primary rat hippocampal neurons, and Jurkat T cells, resulting in down regulation of the catalytic activity and in a reduction of the Ca^2+^/CaM binding ability [85,86]. Moreover, PKA-mediated phosphorylation at Ser74 causes 14-3-3 protein recruitment, thereby blocking Thr108 dephosphorylation to stabilize an inactive form of CaMKKα/1 (Figure 3) [87,88]. In the case of CaMKKβ/2, cAMP/PKA signaling impairs Ca^2+^/CaM-dependent activation, but not its autonomous activity in transfected COS-7 cells, through direct Ser495 phosphorylation (equivalent to Ser458 in rat CaMKKα/1) in the Ca^2+^/CaM binding region. Furthermore, additional Ser100 and Ser511 phosphorylation by PKA mediates the recruitment of 14-3-3 proteins, preventing dephosphorylation of phosphoSer495 and maintaining the inactive form of CaMKKβ/2 [89], consistent with a report demonstrating that 14-3-3γ binding slows down dephosphorylation of PKA-phosphorylated CaMKKβ/2 by protein phosphatase 1 in vitro [90]. In addition to PKA phosphorylation, phosphorylation on three sites (Ser-129, Ser-133, and Ser-137) in the N-terminal regulatory domain (residues 130–152) of human CaMKKβ/2 by cyclin-dependent protein kinase 5 (CDK5) and glycogen synthase kinase 3 (GSK3) reduces the autonomous activity of CaMKKβ/2 to maintain the kinase in a Ca^2+^/CaM-dependent state (Figure 2B,C) [77]. AMPK, a closely proximal downstream kinase for CaMKK, is activated by CaMKKβ/2, immediately phosphorylating an upstream CaMKKβ/2 at multiple sites in vitro, forming a feedback regulatory loop between CaMKKβ/2 and AMPK. Thr144 phosphorylation in rat CaMKKβ/2 (equivalent to Thr108 in rat CaMKKα/1) by the activated AMPK decreases the autonomous activity, converting CaMKKβ/2 into a Ca^2+^/CaM-dependent enzyme [78]. Phosphorylation at multiple sites in CaMKKβ/2 likely disrupts its N-terminal regulatory function to generate an autonomous activity, thereby holding the kinase tightly regulated by Ca^2+^/CaM, in agreement with the finding that CaMKKβ/2–AMPK pathway activation requires Ca^2+^/CaM signaling (Figure 2B,C) [33,34,35]. Analogous to Thr108 phosphorylation in rat CaMKKα/1 by PKA, Thr144 in rat CaMKKβ/2 was phosphorylated by cAMP/PKA signaling in transfected HeLa cells [79] and was dynamically regulated by protein phosphatases [91]. Schumacher et al. reported that death-associated protein kinase-mediated phosphorylation of human CaMKKβ/2 at Ser511 (Ser510 in rat CaMKKβ/2) attenuates Ca^2+^/CaM-stimulated CaMKK autophosphorylation [92], although the effect of Ser511 phosphorylation on CaMKK activity is still unclear. Both CaMKKα/1 and β/2 undergo intramolecular autophosphorylation at multiple sites (Thr93 and Ser179 in rat CaMKKα/1 and Se22, Thr215, Thr482, and Thr517 in rat CaMKK β/2) [22,76]. Particularly, Thr482 in rat CaMKK β/2 is located at the -5 position from Ile477, equivalent to Ile441, an important residue for autoinhibition in CaMKKα/1 (Figure 1C). Under such conditions, CaMKKβ/2 exhibits an increased autonomous activity, caused, at least in part, by autophosphorylation at Thr482, resulting in partial disruption of the autoinhibitory mechanism (Figure 2B). Autophosphorylation of Thr85 in human CaMKKβ/2 induces its autonomous activity, which is disrupted by a T85S mutation [93], an exonic single nucleotide polymorphism (SNP) (rs3817190) in the CaMKKβ/2 gene linked to anxiety and bipolar disorder [94]. It is intriguing to note that Thr85 is conserved only in primates and is replaced by Ala in a rodent enzyme [93].

## 6. CaMKK Inhibitors and Pharmacological Analyses of Signaling Pathways

Protein kinase inhibitors allow for evaluating the physiological significance of target kinase-mediated signaling pathways. In 2002, the first CaMKK inhibitor, 7*H*-benzimidazo-[2,1-*a*]benz[de]isoquinoline-7-one-3-carboxylic acid (STO-609), was developed [95]; it is ATP competitive, cell membrane permeable, and inhibits CaMKKβ/2 activity 5–10 fold more effectively than CaMKKα/1 activity (IC_50_ value = ~1 µM) (Table 2). A mutagenesis study demonstrated that a single amino acid substitution (Val269 in rat CaMKKβ/2 (Val270 in human counterpart)/Leu233 in CaMKKα/1) confers a distinct sensitivity to STO-609 of CaMKK isoforms [96]. This is consistent with the 2.4 Å crystal structure of the catalytic domain of human CaMKKβ/2 complexed with STO-609, indicating that STO-609 forms hydrogen bonds with the backbone atoms of human CaMKKβ/2 Val270 [97]. Similar to other protein kinase inhibitors, STO-609 inhibits some off-target kinases, including casein kinase 2, extracellular signal-regulated kinase 8, and MAPK-interacting kinase 1 [98], also acting as an aryl hydrocarbon receptor agonist [99]. To validate the pharmacological effect of STO-609, we developed a STO-609-insensitive mutant CaMKKβ/2 (Val269Phe mutant), with an IC_50_ value for STO-609 inhibition approximately two orders of magnitude higher than that of the wild type enzyme. Random mutagenesis revealed that Ala292 substitution in rat CaMKKα/1 or Ala328 in rat CaMKKβ/2 by Thr resulted in a 10–100-fold reduction in STO-609 sensitivity [100]. In addition, ionomycin-induced CaMKIV activation in transfected HeLa cells co-expressing CaMKKβ/2 Val269Phe, and ionomycin-induced phosphorylation of AMPK α subunit (at Thr172) in A549 cells stably expressed with FLAG-rat CaMKKβ/2 mutant (Val269Phe, Ala328Thr) were completely resistant to STO-609 treatment, unlike wild type CaMKKβ/2-expressing cells [96,100]. Furthermore, the suppression of axonal outgrowth [101], dendritic development [102], spine formation [103], and inhibition of N-methyl-D-aspartate (NMDA)/glycine-induced ERK1/2 phosphorylation [104] due to STO-609 treatment in rat hippocampal neurons were rescued by an STO-609-insensitive CaMKKα/1 mutant (Leu233Phe) or CaMKKβ/2 mutant (Val269Phe), suggesting that the pharmacological effects of STO-609 in the neurons were likely due to blocking of the CaMKK-mediated signaling pathways. STO-609 has been widely used to examine the roles of CaMKK-mediated signaling in normal and pathophysiological conditions, including protection against prostate and liver cancers [105,106] and nonalcoholic fatty acid disease (NAFLD) [107]. For example, the CaMKK/CaMKI cascade is involved in basal axonal outgrowth and growth cone motility [101], Wnt5a-faciliated axonal outgrowth [108], enlargement of hippocampal dendritic spines [109], activity-dependent synaptogenesis [103], activity-dependent translational initiation [110] in cultured hippocampal neurons, axonogenesis and dendritogenesis in immature cortical neurons [111], macrophage inflammatory response to sepsis [112], and excitation–transcription coupling-mediated vascular remodeling [56]. CaMKK/CaMKIV regulates gene transcription, including glucokinase, ABCA1, and GLUT2 in pancreatic β-cells [113,114,115], and Ca^2+^-induced cofilin phosphorylation by LIM kinase 1 and neurite outgrowth in Neuro-2a cells [116]. Pharmacological inhibition of CaMKKβ/2 with STO-609 impairs the tumorigenicity of liver cancer cells in vivo, possibly mediated by CaMKIV [106], and suppresses CaMKKβ/2-mediated PKB/Akt phosphorylation in ovarian cancer cell lines, resulting in lower cell growth and viability [32]. Anderson et al. showed that the inhibitor directly targets CaMKKβ/2 in vivo, and that it is a useful molecular probe for in vivo CaMKK functional studies by showing the resistance of CaMKKβ/2 null mice to the suppression of food intake [117]. A pharmacological evaluation using STO-609 demonstrated the wide variety of physiological functions of the CaMKKβ/2-regulated AMPK signaling pathway, including glucose-uptake [118,119], T cell antigen receptor-triggering activation in T cells [120], autophagy [121,122], inflammatory response [123], and neuroinflammation [124]. 

As a result of concerns with STO-609′s lack of specificity, small molecular compounds including 3,5-bis(arylamino)-4*H*-1,2,6-thiadiazin-4-one and its analogues (Compound **11**) [125], an orally available CaMKKβ/2 inhibitor (Compound **4t**) [126], GSK650394 (also known as serum- and glucocorticoid-regulated kinase-1 inhibitor) [127,128], and compounds based on scaffold hopping from GSK650394 (SGC-CAMKK2-1) [128] were developed as potent CaMKK inhibitors; they could be used for analyzing the remaining unexplored CaMKK-dependent pathways and the reported effects of STO-609 (Table 2). A novel STO-609-derived CaMKK inhibitor, 2-hydroxy-3-nitro-7*H*-benzo[de]benzo[4,5]-imidazo[2,1-*a*]isoquinolin-7-one (TIM-063), and an inactive analog, TIM-062, lacking a nitro group, were found in a screening using a compound library derived from STO-609 (Table 2) [129]. The inhibitory properties of TIM-063 are similar to STO-609, except that TIM-063 can similarly inhibit both CaMKKα/1 (0.63 µM) and β/2 (0.96 µM). Moreover, TIM-063 has been shown to interact with and inhibit CaMKK in its active state (Ca^2+^/CaM-bound form) but not in its autoinhibited state (Ca^2+^/CaM-unbound form); this interaction is likely reversible, depending on the intracellular Ca^2+^ concentration [130]. TIM-063, but not TIM-062, suppresses the Ca^2+^-induced phosphorylation of AMPK, CaMKI, and CaMKIV in cultured cells. TIM-063, but not TIM-062, attenuated Ca^2+^-induced Ca^2+^-desensitization of the phasic smooth muscle in mouse urinary bladder strips, similarly to STO-609, confirming the involvement of CaMKK in smooth muscle contraction [131]. These results suggest that TIM-063 combined with TIM-062 could be helpful for evaluating the physiological significance(s) of CaMKK-mediated signaling in vivo.

**Table 2 ijms-23-11025-t002:** CaMKK inhibitors.

Inhibitor	Structure	IC_50_ (nM) for		IC_50_ (µM)	Note
		CaMKKα/1	CaMKKβ/2	Cell-Based Assay	
STO-609 [95]	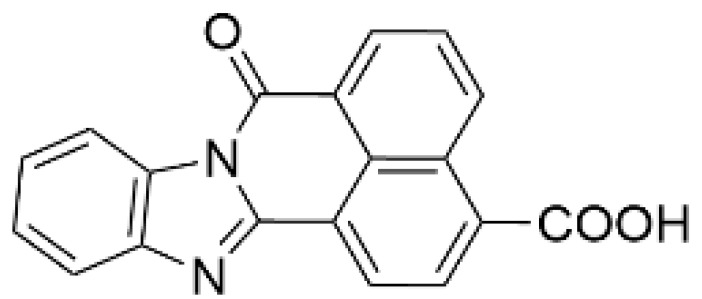	120–408	10–130 [98,128]	0.2	Inhibitor-resistant CaMKK mutants [96,100]
TIM-063 [129]	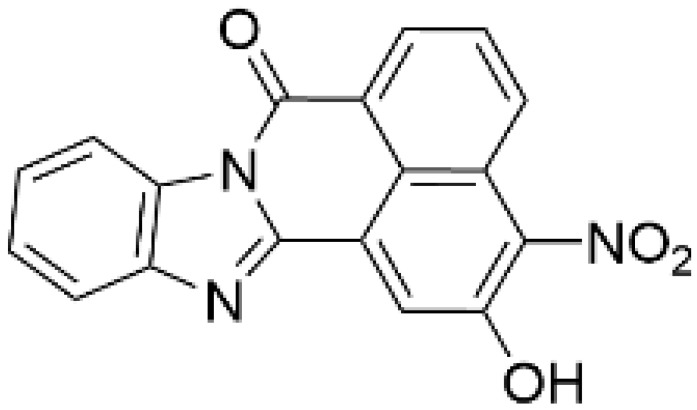	630	960	0.3	Inactive analogue (TIM-062) [129], Conformation-dependent binding [130]
Compound **11** [125]	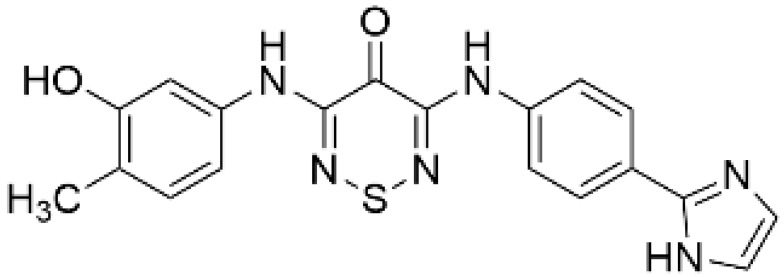	NA	6.5 (µM)	NA	–
Compound **4t** [126]	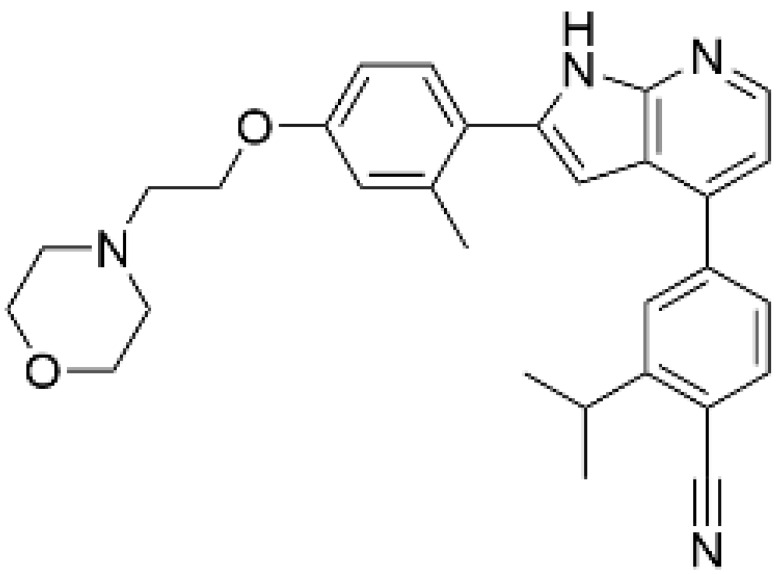	NA	6	NA	Orally available
GSK650394 [128]	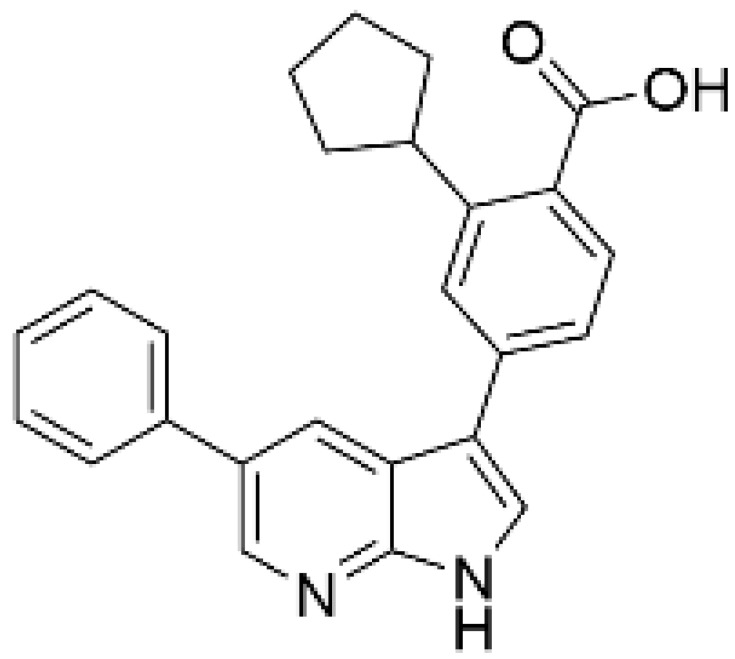	33	3	NA	SGK inhibitor [127]
SGC-CAMKK2-1 ^a^, [128]	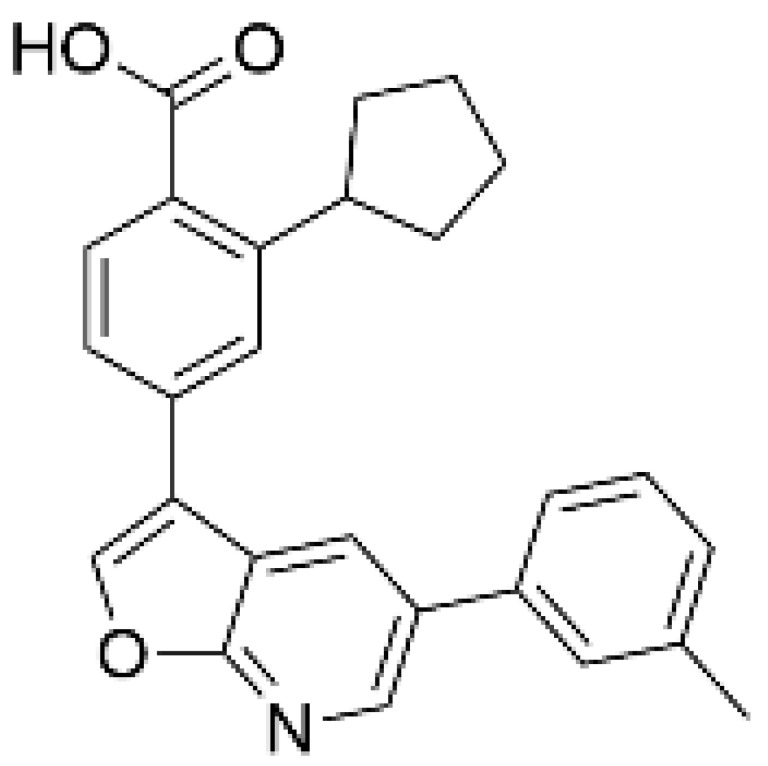	NA	30	1.6	Inactive analogue ^a^ (SGC-CAMKK2-1N)

NA: not available, ^a^
https://www.thesgc.org/chemical-probes/SGC-CAMKK2-1 (accessed on 20 September 2022).

## 7. Genetic Manipulation and Pathophysiological Role of CaMKK

CaMKKα/1 null and hypomorphic mutants [132] and CaMKKβ/2 null mutant mice [133] were generated by Dr. Giese’s group. Both null mutant mice showed normal embryonic and early postnatal development, as well as brain morphology. However, CaMKKβ/2 null mutant mice, in which exon 5 was deleted, showed impaired spatial training-induced CREB phosphorylation (activation) at Ser133 in the hippocampus and spatial memory formation with normal contextual and passive avoidance long-term memory formation [133]. Moreover, CaMKKβ/2 null mutant mice showed impaired long-term, but not short-term, memory for the social transmission of food preferences. Interestingly, the phenotypes caused by genetic CaMKK deletion are sex-dependent. Unlike male mutant mice, female CaMKKβ/2 null mutant mice showed indistinguishable spatial memory formation, hippocampal long-term synaptic plasticity, and CREB phosphorylation levels in the hippocampus from wild type animals [134]. Similarly, CaMKKα/1 null mutant mice, in which exons 4 and 5 were deleted, showed impaired contextual fear memory formation in males but not in females [132]. Blaeser et al. independently showed a defect in long-term contextual fear memory in CaMKKα/1 null mutant mice [135], which correlates with a defect in fear memory in CaMKIV null mice [136]. Both CaMKKα/1 null and hypomorphic mutants exhibited normal spatial memory formation in the Morris water maze [132], suggesting that CaMKKα/1 and CaMKKβ/2 play distinct roles in hippocampus-dependent memory formation. On the other hand, transgenic (Tg) mice expressing a constitutively active form of mouse CaMKKα/1 (residues 1–433) lacking a regulatory domain, including the autoinhibitory and CaM binding sequence [25,137] in the forebrain, also showed impaired spatial memory and contextual fear memory retention with increased basal CaMKI phosphorylation [138]. These effects of constitutively active CaMKKα/1 in Tg mice might be due to the activation of Ca^2+^-independent targets, including PKB/Akt and AMPΚ. These findings also suggest that appropriate levels and timing of CaMKK activation are required for normal neuronal function. CaMKKβ/2 null mice showed decreased food intake and resistance to high-fat diet-induced adiposity, glucose intolerance, and insulin resistance when fed with a high-fat diet, caused partly by a reduced mRNA expression of neuropeptide Y and agouti-related protein, the most potent appetite-stimulating peptides, in the the hypothalamus and unresponsiveness to the orexigenic effects of exogenously administered ghrelin [117]. A specific CaMKKβ/2 reduction in the liver of high-fat diet-fed CaMKKβ/2 (floxed) mice resulted in lower blood glucose and improved glucose tolerance. Hepatocytes from CaMKKβ/2 null mice showed less glucose production and increased de novo lipogenesis and fat oxidation [50]. Consistently, liver-specific CaMKKβ/2 knockout (CaMKKβ/2^LKO^) male mice showed improved glucose tolerance and peripheral insulin sensitivity after 13 weeks of a high-fat diet [37]. Based on studies in genetically engineered-CaMKK null mice, CaMKK-mediated signaling pathways are deeply involved in neuronal plasticity and metabolic regulation (Figure 3).

The CaMKK2/β/AMPK cascade plays important roles in the regulation of the energy metabolism and metabolic processes [117,118,119,120,121,122,123,124]; this role is the same in cancer cell proliferation. CaMKKβ/2 is overexpressed in prostate cancer cells and androgen-dependent CaMKKβ/2 upregulation induces cancer cell growth [105], migration, and invasion [139] via AMPK activation. CaMKKβ/2 is also highly expressed in hepatic cancer cells and the CaMKKβ/2-mediated CaMKIV activation pathway regulates liver cancer cell growth through the mammalian target of the rapamycin/ribosomal protein S6 kinase pathway [106]. SNPs in CaMKK are reportedly associated with various human diseases, including schizophrenia [140]. In addition to an exonic variant, Thr85Ser (rs3817190), in human CaMKKβ/2 associated with behavioral disorders such as anxiety [94], a de novo mutation encoding Arg311Cys, which reduces the CaMKKβ/2 catalytic activity and its apparent affinity for Ca^2+^/CaM [74], was identified in bipolar disorder [141]. In human CaMKKα/1, a variant (rs7214723) causing Glu375Gly substitution was associated with lung cancer [142] and cardiovascular diseases [143], although the effect of the amino acid substitution on CaMKKα/1 function remains to be elucidated. In LKB1-deficient lung cancer cells, α-ketoglutamate, increased by excessive glutamate degradation, binds to and activates CaMKKβ/2 by enhancing CaMKKβ/2 binding to AMPK, conferring resistance to anoikis caused by detachment-induced stress [144]. It is noteworthy that STO-609-treated NAFLD model mice showed a decrease in metabolites associated with catabolic processes and an increase in glycolytic metabolites, suggesting amelioration of nonalcoholic fatty liver with STO-609 treatment [107]. Under ischemic conditions, CaMKKβ/2 exhibits protective roles in the endothelial cells and blood–brain barrier through SIRT1 phosphorylation and activation [145], consistent with a report showing that the genetic deletion of CaMKKβ/2 in female mice exacerbated ischemic injury and increased hemorrhagic transformation after stroke [146].

## 8. Conclusions

Since its discovery, different experimental approaches have demonstrated the importance of CaMKK as a Ca^2+^-dependent regulatory hub of multiple independent signaling pathways mediated by downstream effector proteins, including kinases such as CaMKI/IV, PKB/Akt, and AMPK. In addition, altered CaMKKβ/2 expressions and mutations have been linked to pathophysiological conditions such as multiple cancers and mental disorders. Compared with CaMKKβ/2, the physiological role(s) of CaMKKα/1 has received little attention. Therefore, research of the mechanistic processes underlying CaMKK and downstream target kinases is required in order to shed light on the still unknown physiological and pathophysiological roles of CaMKK-mediated Ca^2+^-signaling, as well as to inform the development of new therapeutic strategies.

## Figures and Tables

**Figure 1 ijms-23-11025-f001:**
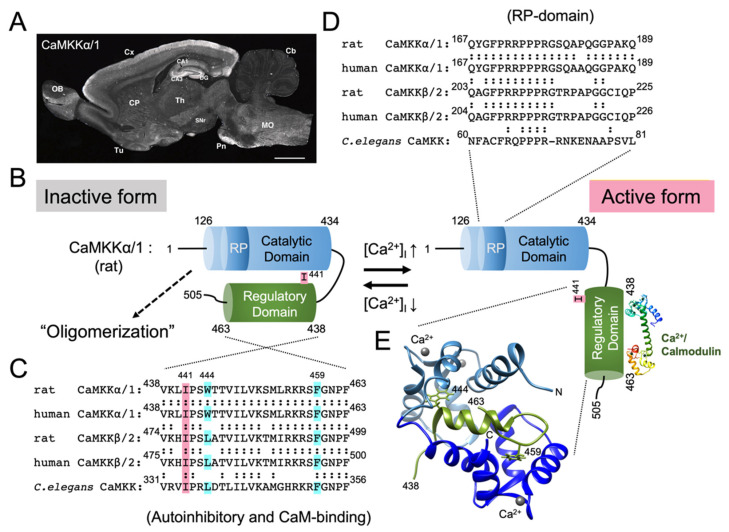
CaMKKα/1; activation mechanism, immunohistochemical localization in the rat brain, and Ca^2+^/CaM-binding. (**A**) Sagittal section of the adult rat brain immunostained with a monoclonal antibody against CaMKKα/1 (reproduced from Ref. [61], with permission from John Wiley and Sons). CA1 and CA3, CA1 and CA3 subregions of Ammon’s horn of the hippocampus; Cb, cerebellar cortex; CP, caudate putamen; Cx, cerebral cortex; DG, dentate gyrus; MO, medulla oblongata; OB, olfactory bulb; Pn, pontine nuclei; SNr, substantia nigra pars reticulata; Th, thalamus; and Tu, olfactory tubercle. Scale bar = 2.5 mm. (**B**) Proposed model of CaMKKα/1 activation mechanism. At low intracellular Ca^2+^ concentration, CaMKKα/1 is in an inactive conformation, where the catalytic domain (residues 126−434) is tightly associated with the regulatory domain (residues 438−463, **C**). With increasing intracellular Ca^2+^ concentration, Ca^2+^/ CaM binds to regulatory domain of CaMKKα/1 (**E**) to suppress autoinhibition, thereby activating the kinase [64]. An activated CaMKK recognizes and phosphorylates downstream kinases including CaMKI, IV, and AMPK by using an Arg/Pro rich insert domain (RP-domain, **D**) [39,65]. Amino acid sequence alignments of the regulatory domain including the autoinhibitory and Ca^2+^/CaM binding segments (**C**) and RP-domain (**D**) in various CaMKKs (rat, human α/1 and β/2 isoforms, and *C. elegans*). Trp(W)444 and Phe(F)459 in rat CaMKKα/1 (**C**) are conserved anchoring residues (indicated by light blue boxes) to the N- and C-terminal hydrophobic pockets of Ca^2+^/CaM, respectively [66]. Ile(I)441 (indicated by a pink box, **C**) is important for rat CaMKKα/1 autoinhibition [64]. (**E**) Ribbon diagram of the NMR structure of Ca^2+^/CaM-CaMKKα/1 regulatory domain peptide (residues 438−463, **C**) complex was obtained from the Protein Data Bank (PDB) entry 1ckk [66] and was visualized using the UCF Chimera [67]. Modified from Ref. [68].

**Figure 2 ijms-23-11025-f002:**
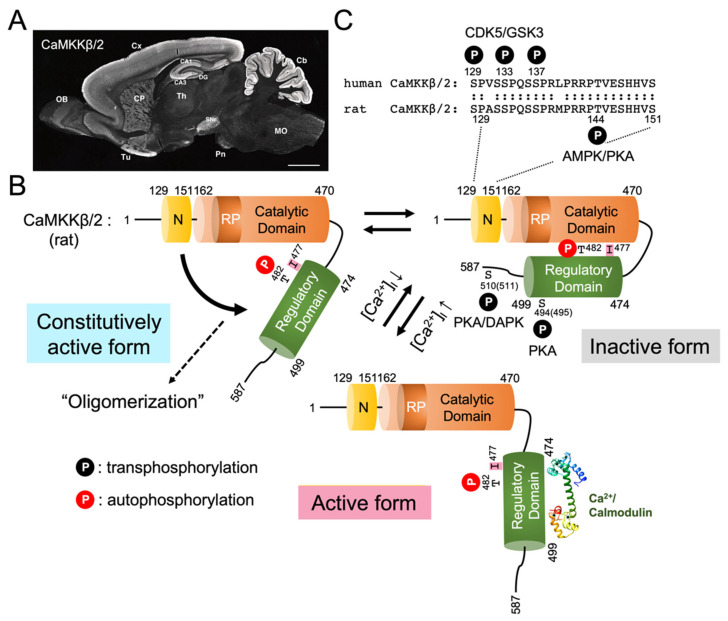
CaMKKβ/2 activation mechanism and immunohistochemical localization in the rat brain. (**A**) Sagittal section of the adult rat brain immunostained with a monoclonal antibody against CaMKKβ/2 (reproduced from Ref. [61], with permission from John Wiley and Sons). CA1 and CA3, CA1 and CA3 subregions of Ammon’s horn of the hippocampus; Cb, cerebellar cortex; CP, caudate putamen; Cx, cerebral cortex; DG, dentate gyrus; MO, medulla oblongata; OB, olfactory bulb; Pn, pontine nuclei; SNr, substantia nigra pars reticulata; Th, thalamus; and Tu, olfactory tubercle. Scale bar = 2.5 mm. (**B**) Proposed model of activation mechanism of CaMKKβ/2. CaMKKβ/2 is constitutively active, exhibiting Ca^2+^/CaM-independent activity (60–70% of total activity), attributable to the N-terminal regulatory segment (residues 129–151, **C**) [22,31]. CaMKKβ/2 exhibits increased autonomous activity, caused, at least in part, by intramolecular autophosphorylation at Thr482, resulting in partial disruption of the autoinhibitory mechanism [76]. Phosphorylation at multiple sites in CaMKKβ/2 by CDK5 and GSK3 [77], activated AMPK [78] or PKA [79], likely disrupting the N-terminal regulatory function to generate autonomous activity, thereby holding the inactive kinase in the absence of Ca^2+^/CaM, in agreement with the finding that CaMKKβ/2-AMPK pathway activation requires Ca^2+^/CaM signaling [33,34,35]. (**C**) Amino acid sequence alignment of the N-terminal regulatory segment in rat and human CaMKKβ/2. CDK5/GSK3 phosphorylate human CaMKKβ/2 at Ser129, Ser133, and Ser137 [77]. Activated AMPK and PKA phosphorylate Thr144 in rat CaMKKβ/2 [78,79]. Modified from Ref. [68].

**Figure 3 ijms-23-11025-f003:**
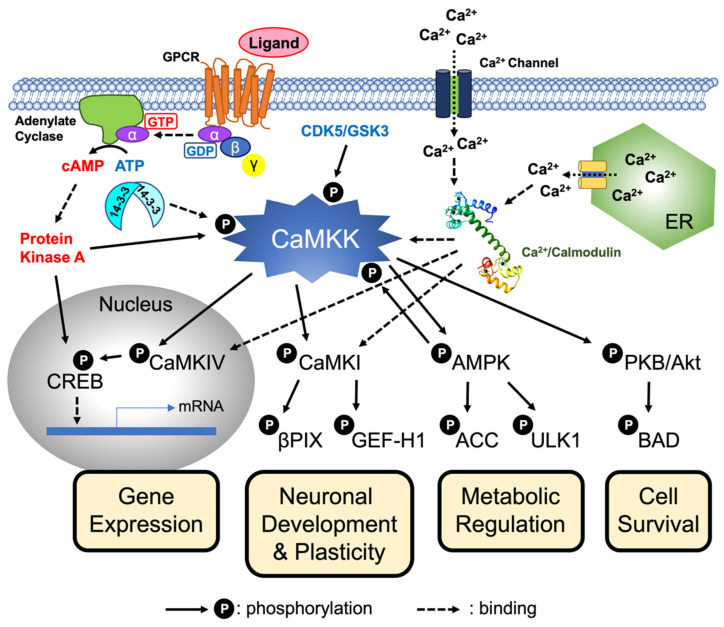
CaMKK-mediated cellular signaling. Increasing intracellular Ca^2+^ concentration triggers the Ca^2+^/CaM-dependent activation of CaMKK, resulting in the activation of the downstream protein kinases including CaM-kinase I (CaMKI), CaM-kinase IV (CaMKIV), AMPK (5′AMP-kinase), and protein kinase B (PKB/Akt) through the phosphorylation of their activation-loop Thr residues. The CaMKK-mediated phosphorylation cascade is involved in a wide variety of physiological functions including transcriptional activation, neuronal development and plasticity, metabolic regulation, and cell survival. CaMKK is regulated by multiple cellular signaling cascades, such as intracellular Ca^2+^, cAMP/PKA signaling, 14-3-3-binding, feedback phosphorylation by activated AMPK, and cyclin-dependent protein kinase 5 (CDK5)/glycogen synthase kinase 3 (GSK3)-mediated phosphorylation. Modified from Ref. [68]. Cream yellow boxes indicate physiological functions of CaMKK-mediated signaling pathways. CREB; cAMP-response element binding protein, βPIX; Pax-interacting exchange factor β, GEF-H1; guanine nucleotide exchange factor H1, ACC; acetyl-CoA carboxylase, ULK1; Unc51-like-kinase 1, and BAD; BCL2 associated agonist of cell death.

**Table 1 ijms-23-11025-t001:** CaMKK in eukaryotic species.

CaMKK		Species	UniProtKB	M.M. (Da)	Ca^2+^/CaM	Substrates
				(a.a. Residues)	-Dependency	(Phosphorylation Site)
CaMKKα/1		rat	P97756	55,908 (505) [20]	YES [25]	CaMKI (α: Thr177) [20,26]
		mouse	Q8VBY2	55,838 (505)		CaMKIV (Thr196) [20,25]
		human	Q8N5S9	55,735 (505)		PKB/Akt (Thr308) [27]
						BRSK1 (Thr189) [28,29]
						Syndapin I (Thr355) [30]
CaMKKβ/2		rat	O88831	64,446 (587) [21,22]	YES [22]	CaMKI (α: Thr177) [22]
		mouse	Q8C078	64,618 (588)	(autonomous activity) [22,31]	CaMKIV (Thr196) [21,22]
	–1	human	Q96RR4-1	64.746 (588) [22]		PKB/Akt (Thr308) [32]
	–2	human	Q96RR4-2	58,899 (533) [23]		AMPK (α: Thr172) [33,34,35]
	–3	human	Q96RR4-3	59,602 (541) [24]		SIRT1 (Ser27, Ser47) [36]
						GAPDH, Pex3 [37]
CKK-1 [38]	–a	*C. elegans*	Q3Y416-2	48,940 (432)	YES [39]	CMK1 (Thr179) [40]
	–b	*C. elegans*	Q3Y416-1	60,804 (541)		
CMKC [41]		*A. nidulans*	Q9Y898	59,153 (518)	YES	CMKB (Thr179) [41]
Ssp1 [42]		*S. pombe*	P50526	73,992 (652)	ND	Ssp2 (Thr189) [42,43]

M.M.; molecular mass, a.a.; amino acid, ND; not determined.

## Data Availability

This review includes partial modifications from [68].

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
