# Peer review of "Molecular Mechanisms Underlying Ca2+/Calmodulin-Dependent Protein Kinase Kinase Signal Transduction"

_ijms, 2022, doi:10.3390/ijms231911025_

Round 1

Reviewer 1 Report

The authors have written a comprehensive and timely review on the CaMKK signaling pathway. It is very readable and the figures are of a high quality. They cover all current topics of interest. This review will be an excellent resource for researchers in the field, as well as for introducing the CaMKK field to the general reader.

Reviewer 2 Report

This review on the molecular mechanisms of calcium calmodulin-dependent protein kinase kinase is very well done. The individual topics are presented in a very comprehensive and detailed manner. The illustrations are well done, the literature has been properly researched and integrated into the manuscript, the sub chapters have been well chosen and accurately discussed. A successful work!

Reviewer 3 Report

This is a very informative and comprehensive review about the role of calmodulin-dependent protein kinase kinase (CaMKK) for the regulation of cellular Ca2+-dependent signal transduction mechanisms. The paper is clearly written and should be accepted after some minor corrections:

1) Citations: Line 54, citations [12-14]. Instead of citation [12] better use  

Ca2+/calmodulin-dependent protein kinase enriched in cerebellar granule cells. Identification of a novel neuronal calmodulin-dependent protein kinase. Ohmstede CA, Jensen KF, Sahyoun NE.J Biol Chem. 1989 Apr 5;264(10):5866-75 which is from the same Lab describing CaMKIV  for the first time.

2) In paragraph 11 two interesting papers should be added: a) Inhibition of calcium/calmodulin-dependent protein kinase kinase (CaMKK) exacerbates impairment of endothelial cell and blood-brain barrier after stroke. Sun P, et al. Eur J Neurosci. 2019. PMID: 30422362

b) Genetic deletion of calcium/calmodulin-dependent protein kinase kinase β (CaMKK β) or CaMK IV exacerbates stroke outcomes in ovariectomized (OVXed) female mice. Liu L, et al. BMC Neurosci. 2014. PMID: 25331941

3) In lines 188-189 the authors state that both CaMKKs "were localized primarily to the perikaryal cytoplasm and dendrites in most immunoreactive neurons (Fig.1A and 2A). This statement is not clearly understood from the figures.

4) A stylistic comment: a) lines 127 and 128 should be moved before line 149 b) The sentence between lines 157 and 164 is too long and should be rephrased. c) Line 239 should be moved directly after line 238 d) Line 300 should be moved directly after line 299  

Reviewer 4 Report

In the manuscript entitled “Molecular mechanisms underlying Ca2+/calmodulin-dependent protein kinase kinase signal transduction”, Tokumitsu and Sakagami present a rather comprehensive review on Ca2+/calmodulin-dependent protein kinase kinase (CaMKK). It is a little difficult to digest. Among the 140 cited references, only less than 30 were published within last 5 years.

Although I like the historical part of this manuscript, it would be nice if the authors could re-organize the review to make the readers easier to digest. I would suggest:

1)     Section 2-Section 6 only contains about 2 citations that were published within last 5 years. They could be condensed a little bit more, or at least be grouped into two sections, with titles like a) The finding and current known members of the CaMKK family; b) Cellular and tissue distribution of CaMKK.

2)     Section 7&8 could be merged into one, with a title like “CaMKK signaling pathway”

3)     Move section 10 forward, maybe after section 6. This section solely discuss pharmacology of CaMKK, leave the pathophysiological effects identified with this pharmacological tools to the last section.

4)     Merge section 9 with 11, and also combine some content from section 10. The scope of this section is to discuss pathophysiological roles of CaMKK signaling.

Minor suggestions:

1)     From the description, it is not very clear to me whether CaMKKβ/2 is constitutively active. If it is, please make it clear in the text, and modify Fig.2B to illustrate this point. Also, please add the recent oligomerization results into Fig. 2B.

2)     In Fig. 3, the illustration of those CaMKK functions look like “yellow highlights” readers usually make in PDF or word files. Please correct them. Also, feed-back phosphorylation of CaMKK by its targets are not illustrated in this figure. Please add this information into Fig. 3.
